# Strong-coupling of WSe$_2$ in ultra-compact plasmonic nanocavities at room temperature

Marie-Elena Kleemann[1], Rohit Chikkaraddy[1], Evgeny M. Alexeev[2], Dean Kos[1], Cloudy Carnegie[1], Will Deacon[1], Alex Casalis de Pury[1], Christoph Große[1], Bart de Nijs[1], Jan Mertens[1], Alexander I. Tartakovskii [2] & Jeremy J. Baumberg [1]

Strong coupling of monolayer metal dichalcogenide semiconductors with light offers encouraging prospects for realistic exciton devices at room temperature. However, the nature of this coupling depends extremely sensitively on the optical confinement and the orientation of electronic dipoles and fields. Here, we show how plasmon strong coupling can be achieved in compact, robust, and easily assembled gold nano-gap resonators at room temperature. We prove that strong-coupling is impossible with monolayers due to the large exciton coherence size, but resolve clear anti-crossings for greater than 7 layer devices with Rabi splittings exceeding 135 meV. We show that such structures improve on prospects for nonlinear exciton functionalities by at least $10^4$, while retaining quantum efficiencies above 50%, and demonstrate evidence for superlinear light emission.

[1] NanoPhotonics Centre, Cavendish Laboratory, University of Cambridge, Cambridge CB3 0HE, UK. [2] Department of Physics and Astronomy, University of Sheffield, Sheffield S3 7RH, UK. Correspondence and requests for materials should be addressed to J.J.B. (email: jjb12@cam.ac.uk)

Transition metal dichalcogenide (TMD) semiconductor heterostructures open up many possibilities for photonics. Of major interest is the strong binding energy of their excitons which allows for room temperature exciton devices[1–3], thus preferable to traditional III–V GaAs-based heterostructures where excitons ionise at 300 K. When such strong excitonic dipoles are embedded in optical resonators, the resulting modified optical density of states changes their emission lifetime through the Purcell factor[4–7]. In extreme cases, when the light–matter coupling is strong enough, the regime of Rabi oscillations and strong coupling is reached, where excitons and photons form new mixed polariton states. These exhibit many compelling features, including Bose condensation[8–10] with superfluid characteristics[11] and a wide range of applications, such as low-energy switching[12,13] or tuneable low-threshold semiconductor lasing[14].

So far, the optical resonators used with TMDs are based on dielectric or metal mirrors, and mostly assisted by cryogenic cooling of monolayers to obtain strong coupling[15,16] with splittings exceeding $k_BT$. At room temperature spectral splittings[3,17–19] are barely resolved and on the order of thermal energies. In order to access many of the key features, clearly resolved polaritons are demanded with splittings greater than 100 meV at room temperature exceeding the plasmon damping of ~90 meV. Since the light–matter interaction depends inversely on the cavity volume, $g \propto 1/\sqrt{V}$, compact optical modes are favoured for nonlinear optical and switching devices. To significantly improve on dielectrically confined light (with minimum volume $V_{min} \sim (\lambda/n)^3$ in refractive index $n$), plasmonic resonators are required[20]. Gap plasmons allow to reach optical volumes below 50 nm$^3$ straightforwardly[20], even below 1 nm$^3$ in certain cases[21]. However, in such extreme plasmonic cavities the resonant optical field is typically polarised perpendicularly to the layer planes[22,23] and hence poorly coupled to the exciton dipole oriented in-plane[24] (see below). This suggests that strong coupling of TMDs in plasmonic cavities is problematic, in comparison to systems with dipoles oriented out-of-plane[25–27]. One approach is to use Type-II stacked TMD bilayers to create dipolaritons, but these are currently challenging to fabricate.

Here, we present a different approach to TMD-plasmon interactions, which is capable of room temperature strong-coupling in ultra-compact resonators with Rabi splittings exceeding $\Omega_R > 140$ meV. We show that the high refractive index ($n \sim 3$) of

TMD layers retunes the plasmons so that thicker gap cavities are required to reach appropriate resonance conditions. Surprisingly, we show that multilayers ($N_L > 7$) of WSe$_2$ achieve this strong-coupling, despite their typically weak luminescence from exciton recombination, which is indirect. We explain this strong-coupling by the dramatic Purcell enhancements in the plasmonic cavities, which competes with carrier scattering so that optical re-emission overcomes phonon-driven intervalley transfers. This localised plasmonic geometry thus provides a route to unusual nonlinear optics (for instance also based on spin/valley-selection) as well as to accessing photon blockade effects at room temperature.

## Results

**Construction of plasmon-coupled TMDs.** Vertically stacked atomically thin materials exhibit a wide range of optical and electrical properties. Semiconductor TMDs, such as MoS$_2$ and WSe$_2$ have strong excitons in the visible and near-infrared (IR) with spin-selective excitation into K,K′ valleys[28,29]. Monolayers become direct-gap and produce strong exciton photoluminescence (PL) which is suppressed as soon as $N_L \geq 2$. In strong coupling, the exciton emission rate has to exceed both the cavity loss rate $\kappa$ and the exciton scattering rate $\gamma_x \geq k_BT$. In this regime, excitons with oscillator strength $f$ give Rabi splitting $\Omega_R \propto \sqrt{fN_Ln/\lambda^3}$. Since the band structure of TMD multilayers changes from that of monolayers when they are van der Waals stacked, we expect that layers must be spaced (for instance with hBN[16]) to retain efficient emission, increasing the cavity volumes and reducing their Rabi coupling.

A second problem is the tuning of the different plasmon modes in nanometre-scale gap cavities. Smaller gaps $d$ result in tighter plasmonic confinement ($V \sim d^2D$ for nanoparticles of diameter $D$); however, the resonant modes rapidly red-shift to the IR, out of resonance with the excitons. This is a particular problem for cube nanoparticle geometries, which have been used to produce enhanced Purcell factors when coupled to semiconductor quantum dots[4] or MoS$_2$[30], but only for gaps $d$ of order 10 nm. We recently showed that truncated spherical nanoparticles (as produced naturally by the equilibrium faceting) are optimal to produce strong coupling in the visible/near-IR[31]. Here, we balance the gap size with the TMD refractive index to achieve optimal strong-coupling conditions.

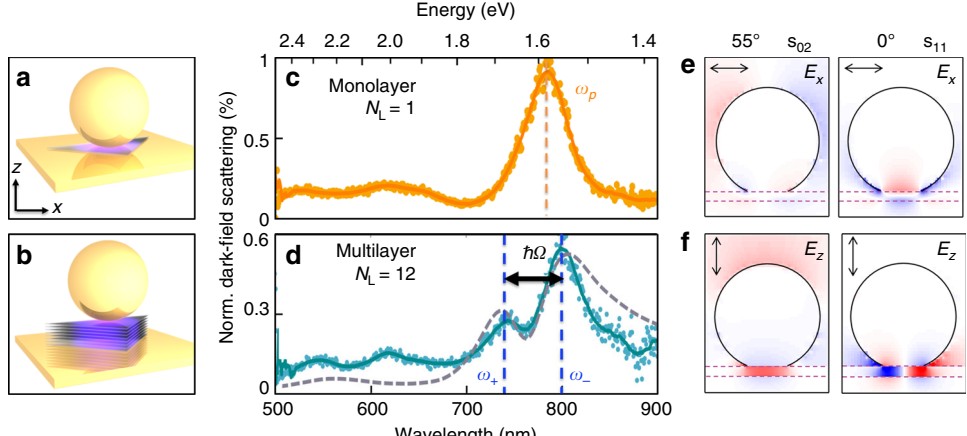

**Fig. 1** Optical signature of single and multilayer WSe$_2$ embedded in plasmonic nanocavities. **a, b** Schematic of nanoparticle-on-mirror cavity encapsulating WSe$_2$ flakes for $N_L = 1, 12$. **c, d** Dark-field scattering of individual NPoMs showing single-plasmon peak for $N_L = 1$ WSe$_2$, but a mode splitting due to strong coupling for multilayer $N_L = 12$ reproduced by FDTD simulations (dashed line). **e, f** Simulated field distributions $E_{x,z}$ for two plasmon-coupled modes (s$_{02}$, s$_{11}$) at $\lambda = 780$ nm with high refractive index material (dashed) in the 10 nm gap ($n = 2.8$). Angles of incidence 55° (left) and 0° (right); red-blue colour-scale shows field enhancements of $\pm 30|E_0|$

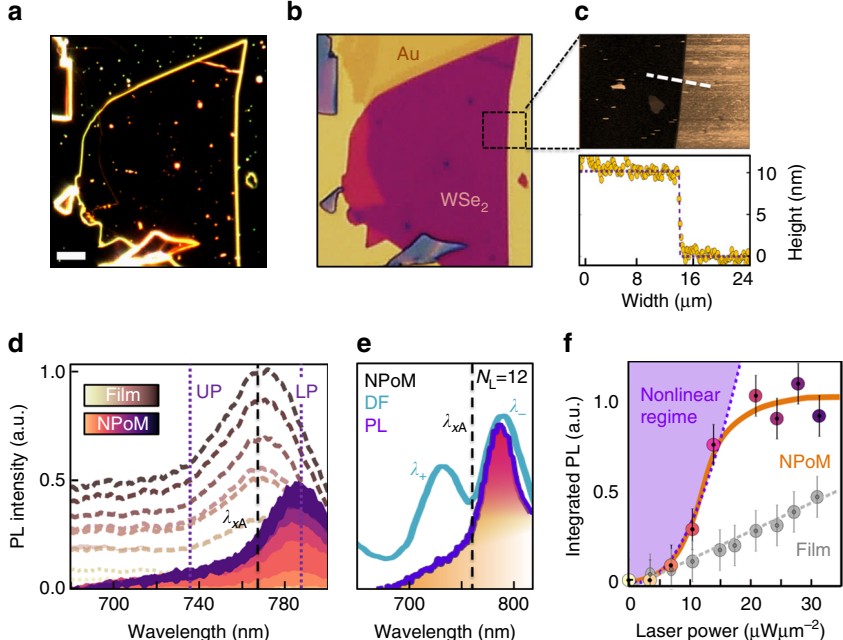

**Fig. 2** WSe$_2$ stack properties. **a** Dark-field image with 10 μm scale bar (solid white line), **b** Bright-field and **c** AFM images of multilayer WSe$_2$ stack. The AFM height profile of flake edge (white dashed line) gives $d = 10 \pm 0.4$ nm, corresponding to 12 layers. **d** Power-dependent emission (Input power ranges from 0 to 30 μW μm$^{-2}$ in steps of 2.5) at room temperature of the multilayer stack (brown) and NPoM (purple). Vertical lines mark A exciton absorption peak (black) and polariton peaks (purple). **e** Scattering and PL emission spectra on same NPoM, dashed line marks exciton absorption peak. **f** Scaling of integrated emission intensity with pump power for polaritons (orange curve) in NPoM (background subtracted) and (grey) excitons in multilayer on Au substrate. Error bars result from experimental noise and are extracted from the standard error of the background counts. Purple shading indicates the nonlinear emission regime

Our self-assembled plasmonic cavities use the nanoparticle-on-mirror geometry (NPoM, equivalent to gap-mode patch antennas) to embed exfoliated WSe$_2$ multilayers of increasing $N_L$ inside the gap. Briefly, TMD layers are transferred to template-stripped Au substrates and Au nanoparticles of different diameter $D = 60$–$100$ nm are drop-casted on top to form NPoMs. These are spaced far enough apart ($>5$ μm) to observe them individually. Image charges induced in the underlying Au film plasmonically couple each facetted nanoparticle with its mirror image, enclosing a nano-cavity and creating a system resembling the coupled plasmon dimer (Fig. 1a, b). Resonance with the A exciton is achieved by tuning the cavity resonance for different gap thicknesses by selecting $D$. White-light dark-field (DF) microscopy, with light incident at high angles ($|\theta| > 55°$) and collected at lower angles, is then used to identify the plasmonic resonances (Fig. 1c, d) and resolve the hybrid plasmon–exciton branches for single NPoM cavities.

For $D = 100$ nm NPoMs, the dominant coupled plasmon resonance $\lambda_p$ in the system is beyond 700 nm (shorter $\lambda$ resonances correspond to transverse and quadrupoles with weaker field confinement and enhancements), set by the exact geometry[32]. In these nanocavities, field enhancements up to $|E|/|E_0| \approx 70$ inside the high $n$ layer (Fig. 1e) should yield Purcell factors sufficient to produce strong coupling[33]. However, incorporating $N_L = 1$ MoS$_2$ or WSe$_2$ monolayers in such nanogaps, and observing them by using dynamic tuning of the plasmons (Supplementary Fig. 1) always gives weak coupling only, merely enhancing PL yields by up to 5-fold when on resonance (note off-resonance also enhanced in such gaps).

**Strong coupling in WSe$_2$ flakes**. By incorporating thicker WSe$_2$ flakes, we clearly observe the appearance of strong coupling (Fig. 1d). Instead of a single-plasmon mode, a splitting between two exciton–plasmon-polaritons (or plexcitons) is seen, with

peak separations between $\omega_+$ and $\omega_-$ exceeding 140 meV. In this case, an exfoliated flake of thickness 10 nm with $N_L = 12$ layers spaces the Au plasmonic gap (Fig. 2a, b). This generates strong IR plasmonic resonances (appearing red in Fig. 2a) as compared to the NPoM systems produced off the flake which appear green. Atomic force microscopy (AFM) confirms the multilayer thickness (Fig. 2c), which gives a reflection spectrum matching that expected from the dielectric function of WSe$_2$ (Supplementary Fig. 2) and appearing purple in bright field. At room temperature, the large exciton binding energy (400 meV) in WSe$_2$ results in stable excitons with PL from the film region at the A exciton near 761 nm, which agrees with literature[28, 29, 34–36] (Fig. 2d). In comparison, emission from the NPoM (after subtracting background emission from the surrounding film) is shifted to longer wavelengths, emerging instead at the lower polariton peak at 785 nm (Fig. 2d).

**Tunable plasmonic cavity with high Purcell factor**. TMDs switch from direct to indirect band gaps when moving from single-layers to multi-layers[35, 36]. Here we show that by matching the plasmon resonance to the direct A exciton transition in multilayers, we resonantly enhance the radiative decay process at the K,K′ points, speeding it up by more than an order of magnitude. Although in multilayer WSe$_2$ the direct K-K exciton is at 1.63 eV, far above the Γ–K transition energy at 1.4 eV (Supplementary Fig. 3)[35], intervalley relaxation requires phonon interactions that take only about 30 fs at 300 K[37]. Embedding the WSe$_2$ in plasmonic cavities with very high Purcell factor and tuning the plasmon resonance to match the A exciton transition selectively, enhances their radiative recombination (below their undressed radiative recombination time of ~200 fs[37]), thus switching on the strong coupling even for multilayer WSe$_2$.

The clear presence of the flake sandwiched within the plasmonic gap is shown by surface-enhanced Raman scattering

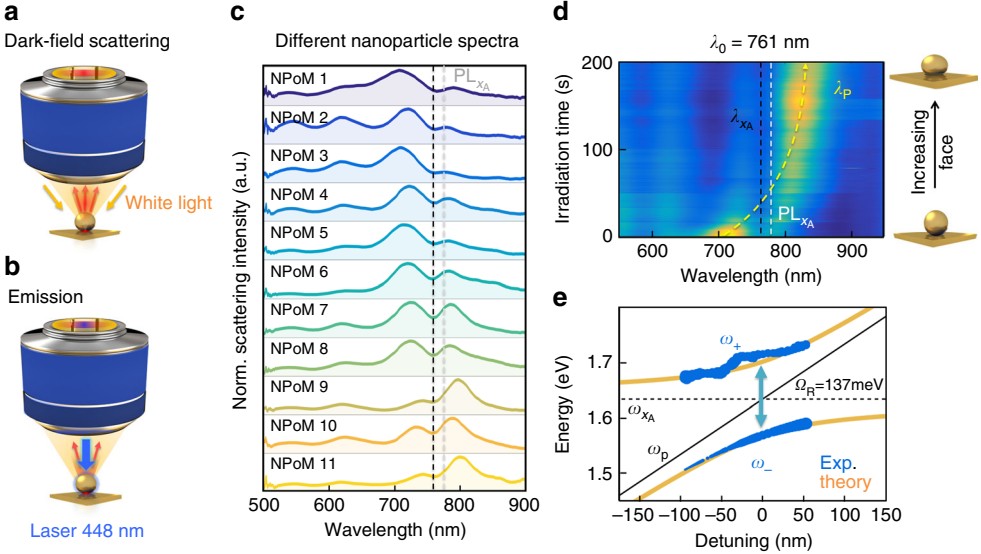

**Fig. 3** Signature of strong coupling in individual NPoM constructs. **a** White-light dark-field scattering, and **b** PL/irradiation setups. **c** Normalised dark-field scattering spectra of individual NPoM constructs each with $N_L = 12$ WSe$_2$ spacer, black dashed line at A exciton position. **d, e** Laser-induced tuning of NPoM plasmon resonance via growth of lower facet, mapping out anti-crossing around A exciton. **e** Hybrid plasmon–exciton branches (plexcitons, $\omega_\pm$) with Rabi-spitting $\Omega_R = 137$ meV. Blue dots correspond to extracted experimental values and orange to coupled oscillator model

(SERS) of the assembled NPoMs. When pumped at 633 nm, typical Raman modes of bulk WSe$_2$ are observed (Supplementary Fig. 4), showing that it remains intact and is not chemically modified by the interaction with the Au or by laser irradiation. Comparing the DF scattering spectra with the PL spectra on the same NPoM (Fig. 2e) shows the latter is red-shifted from the bare exciton emission and absorption, matching instead the lower polariton. As typical for strongly coupled systems, little emission is seen from the upper polariton indicating fast energy relaxation, while the relative emission strength on and off the NPoM suggests PL enhancements exceeding 100, suggesting the speedup of emission compared to non-radiative scattering (see below). Strong-coupling thus confirms that re-emission from the A exciton happens here before phonon scattering can occur. Neither exciton nor polariton lines are observed to spectrally shift with increasing laser power. However, we find that the light emission is superlinear from the NPoM while linear from the surrounding multilayer film (Fig. 2f), providing a first indication of nonlinear polariton effects (see below). The saturation seen at higher powers has been previously attributed to exciton–exciton Auger processes[38].

**Evidence of strong coupling.** DF scattering spectra are measured on many individual NPoM constructs with $N_L = 12$ layers of WSe$_2$ as a spacer material (Fig. 3). The different normalised scattering spectra (Fig. 3c) all show consistent double-peaked spectral features with a characteristic dip at the position of the A exciton. The slight tuning of the plasmon modes arises from variations in the NP sizes and facets[39], which in the strong-coupling regime then produces different relative peak heights as observed. No Fano-like lineshapes are seen, corroborating the regime of strong coupling[40]. By contrast, for WSe$_2$ monolayer constructs ($N_L = 1$), which scale down both the total oscillator strength and the cavity volume, no such dips are seen.

Plotting different spectra from NPoMs traces out the anti-crossing, but we can also tune a single NPoM coupled plasmon resonance so that it tunes across resonance with the A exciton. To do this, we use blue irradiation of the NPoM (0.1 mW of a 448 nm CW laser) to drive Au atoms towards the NP bottom facet, which red-tunes the plasmon, $\omega_p$ (Supplementary Figs. 1,5–7) as

previously studied[41]. The robust WSe$_2$ layers prevent any metal-bridging from NPs onto the bottom mirror, while SERS confirms that the WSe$_2$ remains intact throughout. A clear anti-crossing is also seen in this case (Fig. 3d, e), which can be fit to the semi-classical coupled oscillator model[20, 42] giving

$$2\omega_\pm = (\omega_p + \omega_{xA}) \pm \sqrt{\Omega_R^2 + \delta^2}, \qquad (1)$$

where $\delta = \omega_p - \omega_{xA}$ is the detuning (also extracted from the fits), and each plasmon resonance position (black line) is extracted from the two peak positions. Comparing this oscillator model (orange) with the experimental results (blue) shows good agreement (Fig. 3e). The two hybrid plexcitons exhibit the characteristic plasmon–exciton dispersion with Rabi splitting of 137 meV. The same Rabi splitting is found for all NPoMs on the same number of WSe$_2$ layers (Fig. 3c), showing the consistency of the cavity volume and oscillator strength at different spatial locations, and the robustness of this effect. We also note that $\Omega_R > (\gamma_x + \kappa)/2$ for larger $N_L$ as required for strong coupling (see below), and this model correctly tracks the crossover in linewidths (Supplementary Fig. 8).

**Plasmon mode tuning.** To better understand the system, we use finite-difference time-domain (FDTD) simulations to calculate the mode volume $V$ and the Purcell Factor $F_P \propto Q/V$ (Fig. 4). Because TMDs are highly anisotropic (with non-resonant $n$ which is 50% times smaller along $z$ compared to in-plane), their mode tuning depends sensitively on field orientations (Figs. 4b and 1e). With their large out-of-plane refractive index $n_z = 2.8$,[34] the lowest gap plasmon is red-shifted to $\lambda \approx 800$ nm even for these large 10 nm gaps (Fig. 4b). The two resonances seen when exciton contributions are switched off (leaving just the non-resonant $n$) have different polarisation selection rules, and are seen either at high angles ($z$, s$_{02}$) or normal incidence ($x$, s$_{11}$), according to their symmetry (Figs. 4b and 1e).

These symmetries become crucial when considering not point dipole emitters, such as molecules[20], but spatially delocalised

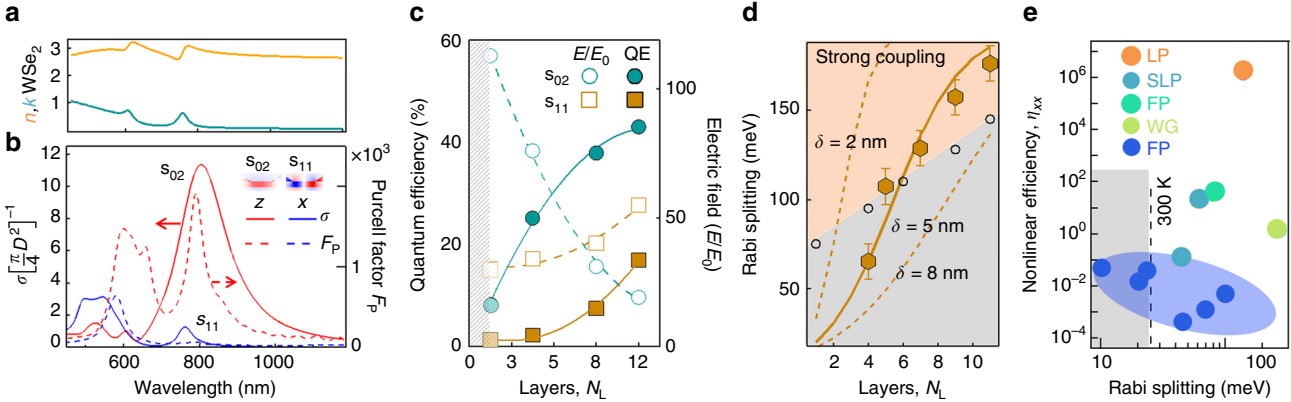

**Fig. 4** Comparison of calculated Purcell and pumping factors. **a** Effective refractive index $n+ik$ of multilayer WSe$_2$. **b** FDTD simulations of the scattering cross section (solid lines) and Purcell factors (dashed) for optical fields along normal ($z$) and in-plane ($x$) directions. **c** Quantum efficiency (solid lines) and field enhancement (dashed) vs. number of monolayers, when keeping s$_{02}$, s$_{11}$ tuned to A exciton by changing the NP diameter (Supplementary Fig. 9). **d** Rabi splitting for NPoMs containing different numbers of WSe$_2$ layers, with fit to model (orange line, see text) depending on exciton interlayer coupling length $\delta$. Error bars originate from the variation in the dark-field scattering and are extracted from the standard error of the fit. Grey circles are obtained from measured plasmon linewidths (NPoM size changed to retain exciton resonance), the strong-coupling regime is shaded orange. **e** Comparison of strong coupling Rabi splittings and exciton nonlinear efficiencies for different cavities: localised plasmon (LP, present work), surface lattice plasmons (SLP), Fabry–Perot mirror cavities (FP, blue=monolayer, green=multilayer), waveguide (WG) (for references see Supplementary Table 1)

excitons. The exciton coherence diameter $d_c$ is given by[43]

$$d_c = 8\hbar\sqrt{\frac{\pi}{M}}, \qquad (2)$$

where the total exciton mass $M = m_e^* + m_h^* = 0.63 m_e$[44] and the exciton homogeneous spectral linewidth at room temperature is $\Delta = 40$ meV[37], resulting in $d_c = 24$ nm for WSe$_2$. This means that the s$_{11}$ excitation (in Fig. 1f) is suppressed since it tries to drive dipoles that are only 20 nm apart with opposite phase within the same exciton. For this reason in our NPoM system, the s$_{02}$ mode dominates exciton coupling (Supplementary Fig. 1).

The Purcell factors here reach $F_P \lesssim 2000$ near 800 nm, speeding up radiative decay from the K,K′ excitons sufficiently to allow strong coupling before intra-valley scattering occurs. While the $E_x$ field strengths aligned to the dominant in-plane exciton[45] dipole orientation are insufficient to retrieve strong coupling, in multilayers the exciton has an out-of-plane component. Using the in-plane refractive index $n_x(\lambda)$ constrained by fitting the reflectivity of the flake away from NPoMs (Supplementary Fig. 2), we estimate that this exciton dipole strength along $z$ is of order 25% of the $x$-dipole for multilayers. Such mixing likely originates from multilayer-induced mixing (evidenced in polarisation-dependent in-plane reflectivity measurements[34]), as stress-induced band mixing is not evident (the SERS phonon lines are found to be unshifted).

For monolayer plasmon cavities, several factors prevent strong coupling. The exciton dipole is now completely in-plane and cannot couple to the strong $E_z$ fields (which contrasts with plasmon lattice modes possessing much larger volumes for which dipolar coupling to $E_x$ works[46]). Second, the much smaller monolayer gap red-shifts the fundamental plasmon resonance far beyond the exciton into the IR, requiring plexciton coupling through higher-order plasmon modes. Since these possess oscillatory fields in the gap (Supplementary Fig. 9), they also cannot couple to the delocalised exciton because of destructive interference as described above, despite their strong field enhancements. If instead monolayers are placed inside wider plasmon gaps to retune the lowest plasmon back to the exciton resonance[30], the larger cavity volume contains too little exciton

oscillator strength (about 30 meV) to reach strong coupling, although this is still significant. Alternatively, reducing the nanoparticle size to tune the plasmon to match the exciton resonance, gives also too small a field enhancement (Fig. 4c). The optimal number of WSe$_2$ layers thus depends on the largest field strength that can be obtained while tuning the lowest plasmon mode into exciton resonance (Fig. 4c).

This can be directly observed by tracking the Rabi splitting for different numbers of WSe$_2$ layers (Fig. 4d; Supplementary Fig. 10, Supplementary Note 7), where the NP size is chosen to keep the plasmon on resonance to the exciton. For thicker multilayers, the larger gap size thus requires a larger NP diameter, which in turn increases the radiative coupling (grey circles). For these larger linewidths, the strong coupling threshold increases, resulting in a restricted region (orange) where it is possible. In this model we assume the out-of-plane exciton dipole $\mu_z$ increases from 0 to 20% of the in-plane dipole, over a vertical coupling distance $\delta$ set by interlayer coupling. We find the best fit to our data when the interlayer coupling delocalises excitons between planes to give a $z$-dipole contribution in multilayers over 5 nm thick (solid line, Fig. 4d), larger than the in-plane 2D exciton Bohr radius (1.5 nm) by a factor similar to the ratio of 3D to 2D exciton Bohr radii in III–V semiconductor quantum wells.

Finally, we comment on the comparison with dielectric-based TMD strong coupling, focussing on how exciton–exciton nonlinearities will work in each system, by identifying an appropriate figure of merit. While the Rabi energy is one key parameter, the nonlinear properties of such systems depend on the exciton density because the underlying Coulombic-interaction depends on the exciton density squared. This scales with the spatial overlap of the confined optical field with the excitonic material component, set by what fraction of each cavity round trip the light spends as an exciton. Given optimal tuning (as in Fig. 3c–e), this nonlinear scaling is controlled by both the cavity field enhancement and the fraction of the light that is inside the semiconductor layers, $F_X = V_X/V$, where $V_X$ is the volume of the optical mode which overlaps with the TMD layers (see Theory). For reports in the literature of TMD strong coupling (Supplementary Table 1), we can thus map this exciton nonlinear efficiency, $\eta_{XX} = (F_P F_X)^2$, which varies by more than 8 orders of

magnitude in different constructs (Fig. 4e). Within the plasmon cavity volume for $N_L = 7$, the exciton Bohr radius indicates there are only 5 non-overlapping excitons (Supplementary Fig. 10d) giving good prospects for exciton nonlinearities. The extreme confinement within the plasmonic cavity here, and its complete filling with TMD layers, thus produces a system which is extremely favourable for nonlinear device performance.

## Discussion

We have successfully shown how ultra-compact plasmonic resonators in combination with TMDs are capable of reaching the strong-coupling regime at room temperature with Rabi splittings exceeding $\Omega_R > 140$ meV. Drastic Purcell enhancement of plasmonic cavities dominates over carrier scattering and phonon-driven intervalley transfers. The simple tuneability of the NPoM geometry allows for optimal coupling in extremely compact cavities with ultrasmall mode volumes. Three considerations are crucial: the coherence size of the exciton compared to the plasmonic mode, the orientation of the exciton dipole which is exactly in-plane in monolayers, but 25% out-of-plane in multilayers, and the radiative coupling which needs to be large enough without tuning the plasmon far into the IR. Our success opens up new routes to create low-energy photonic devices, such as low-threshold lasers and optoelectronic all-optical circuit elements, including polariton switches, transistors and logic gates at room temperature.

## Methods

**Sample fabrication**. Monolayer and mulitlayer TMD flakes are mechanically exfoliated from bulk crystals onto a silicon substrate coated with a polymer bilayer (100 nm polymethylglutarimide (PMGI) and 1 μm polymethyl methacrylate (PMMA)). The top PMMA layer with a chosen crystal is lifted off the substrate by dissolving the sacrificial PMGI layer and is transferred onto a gold substrate. The PMMA membrane is dissolved in acetone and the sample is rinsed in iso-propanol and dried with a nitrogen flow.

Gold substrates are fabricated by thermal evaporation to give a 100 nm Au layer on silicon wafers. Nanoparticles (60–100 nm citrate stabilised Au, BBI Scientific) are assembled directly onto Au substrates covered at low density with WSe₂, MoSe₂ or MoS₂ flakes. Remaining colloidal particles are removed by rinsing with deionised water, and the samples is dried with nitrogen gas.

**Dark-field and emission spectroscopy**. A customised DF microscope (Olympus BX51) is used to perform white-light DF and emission spectroscopy of individual NPoMs. An incandescent light source is focused with a ×100 DF microscope objective providing high angle illumination up to 69° (NA = 0.93). The numerical aperture to collect the scattered light is NA = 0.8 (Fig. 3a). Using a confocal geometry in collection, the scattered light is collected with a 50 μm-diameter fibre as a pinhole to limit the collection area on the sample (1 μm diameter). Spectra are recorded with a cooled spectrometer (Ocean Optics QE65000) with integration times of 1000 ms.

For emission spectroscopy, individual NPoMs are illuminated with a single-mode fibre-coupled diode laser (Coherent CUBE) at 448 nm pump wavelength. Collimated laser light fills the back-focal-plane of the microscope objective, thus illuminating in bright-field a diffraction-limited area of 360 nm diameter on the sample. In order to avoid damage the particles are irradiated with powers below 0.1 mW (corresponding to power densities <0.4 mW μm⁻²) at the sample. The emitted light is recorded with the same spectrometer as for the white-light spectroscopy, with the laser light blocked by a 500 nm long-pass filter (Thorlabs) in the collection path.

**Theory**. Full-wave FDTD simulations are performed to calculate the plasmon cavity resonances for a 100 nm spherical gold nanoparticle (AuNP) with 40 nm bottom facet diameter (as all nanoparticles are facetted[47, 48]). AuNPs are placed above the 100 nm-thick Au mirror with a $d = 1$–$10$ nm gap. The wavelength-dependent refractive index of the gap is extracted from the experimental reflectivity spectrum of the same WSe₂ flake (Fig. 4a) and these parameters are incorporated into full-wave calculations to model the NPoM cavity. The Au NP is illuminated with an s-polarised or p-polarised plane wave (Total-field scattered-field source) from an angle of incidence of $\theta_i = 55°$ or $0°$ (which are known to couple to z or x, y cavity modes). The inbuilt sweep parameter was used to sweep the incident wavelength from 500 to 900 nm. The scattered light for each wavelength is collected within a cone of half angle $\theta_c = 55°$ based on the numerical aperture of the

objective. Since the cavity resonance is sensitive to the facet size, this is tuned to scan the plasmon across the exciton resonance.

To estimate the mode volume of the cavity resonance, the Purcell factor is calculated,

$$F_P = \frac{3}{4\pi^2}\left(\frac{\lambda_0}{n}\right)^3 \frac{Q}{V} \qquad (3)$$

This Purcell factor describes the enhanced spontaneous decay rate of a classical dipole in the NPoM cavity, and is simulated by placing dipoles in specific orientations either in the centre of the facet or at the edge, near the top nanoparticle where the fields are most strongly enhanced (Figs. 1e and 4). From this, and the Q-factor extracted from the resonant lineshape of each mode, the volume was then extracted.

To simulate the strong coupling, the gap of NPoM cavity is modelled as a self-consistent dispersive medium with excitonic dielectric permittivity described by the Lorentz model as

$$\varepsilon_{tot}(\omega) = \varepsilon_\infty + \sum_j \frac{f_j \omega_j^2}{\omega_j^2 - \omega^2 - i\gamma_j\omega} \qquad (4)$$

where $\varepsilon_\infty = 5$ is the non-resonant background, and for each exciton (A, B, C) $f_j$ is the reduced oscillator strength. $\hbar\omega_{A,B} = 1.63$, $2.0$ eV are the A, B exciton energies with linewidth parameters $\gamma_{A,B} = 70$, $100$ meV. The $f_j$ are tuned to match the experimentally observed reflectivity and strong-coupling spectra. This permittivity has a 50% reduced background refractive index for fields normal to the layer planes, with the anisotropy in $f_j$ matched to the experimental data.

The exciton nonlinear response depends on local exciton Coulomb interactions, and thus is proportional to $n_X^2$ (for exciton density $n_X$). The light intensity in the cavity is proportional to Q, giving $n_X \propto \frac{Q}{V_x} = \frac{Q}{V}\frac{V}{V_x} = F_P F_x$ with $F_x = V/V_x$. A suitable figure of merit is then $\eta_{XX} = (F_P F_X)^2$.

**Data availability**. All relevant data present in this publication can be accessed at: https://doi.org/10.17863/CAM.13084.

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

## Acknowledgements

We acknowledge support from EPSRC grants EP/G060649/1, EP/L027151/1, EP/G037221/1, EPSRC NanoDTC and ERC grant LINASS 320503. J.M. acknowledges support from the Winton Programme of the Physics of Sustainability. R.C. acknowledges support from the Dr Manmohan Singh scholarship from St John's College, University of Cambridge. AIT and EMA acknowledge support from EPSRC grant EP/M012727/1, Graphene Flagship grant 696656 and ITN Spin-NANO 676108. CC acknowledges support from the UK National Physical Laboratory. C.G. acknowledges support by the A. v. Humboldt Foundation.

## Author contributions

Experiments were planned and executed by M.-E.K., W.M.D. and C.C. with support from A.C.d.P., C.G., B.d.N. and J.M. Simulations were performed by R.C., while AFM measurements were performed by D.K. The samples were prepared by E.M.A. The data were analysed by M.-E.K., R.C., A.I.T. and J.J.B., and all authors contributed to the manuscript.

## Additional information

**Competing interests:** The authors declare no competing financial interests.

