## [Peer Review File · Nature Communications]

Reviewers' comments:

Reviewer #1 (Remarks to the Author):

Kleemann and coauthors present strong coupling of WSe₂ in a nanometric plasmonic resonant cavity. A very interesting and original observation, which in itself merits consideration of publication in a journal like Nat.Comm.

The methodology is based on the "NPoM" nanoparticle on mirror geometry, the same effective approach also used by the group in refs 20 and 21 to achieve strong coupling on a single organic molecule and extremely enhanced Raman in picocavities. In the current manuscript, the NPoM geometry is, to my knowledge, for the first time applied to a 2D-TMD semiconductor.

The results are at first sight counterintuitive: the strong coupling is not observed for a 1 nm WSe₂ monolayer in the NPoM gap; while a strong coupling of 135 meV does occur for a 10 nm 8-layer WSe₂ in the gap. The observation is surprising, as the smallest cavity would exert the strongest coupling. Yet the monolayer is direct gap and x oriented, whereas the vdWaals stacked multilayers are indirect gap and more z oriented. In addition, the multilayers are less red-shift and thus more in resonance. In the major part of the manuscript authors focus on proving and explaining their proposed understanding.

Authors state that the multilayer with its high refractive index reduces the extreme red-shift of the nanogap, such that the plasmonic resonance (and high Purcell factor) coincides with the direct A-exciton of the multilayer favoring the coupling. It is a peculiar coincidence that the balance between expanding volume (reduced coupling) and resonant tuning (enhanced coupling) should find its optimum at 8 multilayers, exactly the case the authors measured. Figure 4c seems to indicate enhanced Purcell factors for z-orientation even beyond 8 layers, however the reverse for x-orientation. So is the optimum at 8 or beyond? Why did authors not vary the multilayer thickness experimentally? Data on different thicknesses would certainly strengthen the argumentation.

The direct experimental comparison of a monolayer and a bilayer should be most instructive, as volume and resonance are very similar while the difference in WSe₂ bandstructure would directly dominate. Is strong coupling predicted for the bilayer case?

The strong coupling is clearly observed in the double-peaked dark-field spectra on the multilayer WSe₂ constructs. Yet on the exact same system, the A-exciton luminescence is a single fixed peak without any double-peak signature. Why? Is this due to x-orientation, perpendicular to the gap field, of the A-exciton?

Abstract reads: "Strong coupling is impossible with monolayers" and indeed a single plasmon peak is seen. Yet comparing 8 and single layer, the volume V changes around 10x while moving out-of-resonance changes the enhancement by $\sim Q$. Off-resonance balances cavity volume. Thus Q/V is roughly similar for both 8 and single layer. Then the in-plane excitation dipole orientation for the monolayer must be the main reason for the weaker coupling? Or is the delocalization of the exciton even more important? Could authors quantify these factors and estimate the resulting coupling for the monolayer case. Is it still in the meV range or really much lower??

Figure 1e and 1f. It would be clearer to swap 1e and 1f such as to have the x orientation side-by-side to the monolayer and the z with the multilayer.

Reviewer #2 (Remarks to the Author):

The authors report on the observation of room temperature strong coupling in a nanoplasmonic architecture with a multilayer WSe₂ stack. They successfully map out the strong coupling regime by measuring various devices with different dimensions (which I find not convincing at all!), and also, by dynamically tuning the cavity with a laser (Which I find much more convincing!). As far as I can judge, the assumption that the device is operated in the strong coupling regime, is well justified.

Here, I have one general remark, and a few technical remarks:

General: The authors' plasmonic device, operated in the strong coupling regime with a single emitter, has recently been published in Chikkaraddy, R. et al. Single-molecule strong coupling at room temperature in 409 plasmonic nanocavities. *Nature* 535, 127–130 (2016). The room temperature strong coupling with a plasmonic device with 2D materials has been discussed in Liu, Wenjing, et al. "Strong Exciton–Plasmon Coupling in MoS₂ Coupled with Plasmonic Lattice." *Nano letters* 16.2 (2016): 1262-1269. While I think that the paper by the authors is elegant, due to the above mentioned two reports (and the absence of new, other physics), I think it's suited in a more specialized journal.

Technical comments:

1) Ref. 14 does not seem to be the most appropriate reference in the context of 'low threshold polariton lasers'

2) When the authors write 'often spectral splittings^{17–19} are barely resolved and are on the order of thermal energies', do they want to give credit that in ref 17-19 spectral splittings were observed, or do they want to express their criticism about the poor visibility about the latter? I think this formulation is very indirect...

3) I believe the statement "maximising the Rabi coupling, $\Omega = \sqrt{g^2 - (\gamma_x - \kappa)}$ by minimising $\gamma_x - \kappa$ "^{19,42–44}

is not fully correct. The measurable splitting (in reflection, absorption, PL or transmission) follows different formulas which are all detailed in

Savona, Vincenzo, et al. "Quantum well excitons in semiconductor microcavities: Unified treatment of weak and strong coupling regimes." *Solid State Communications* 93.9 (1995): 733-739.

This also makes the next statement about the 'visibility' unnecessary. As the authors imply with this statement, increasing cavity damping to compensate emitter damping does not work...

4) As a side remark, it is interesting that the paper by Liu et al. *Nature Materials* (2015) is only cited in this context, rather than in the introduction as the first claim of strong coupling with a monolayer of MoS₂. I think either the authors should give full credit to the paper as the first strong coupling demonstration in TMDCs, or fully remove the article from the reference list.

5) I think the discussion about the nonlinearity parameter is of pedagogical value, but not backed by any experimental data as far as I can see. As such, it does not have significant value to the paper in my opinion.

6) I could not track the references in Table 8.

7) The concept of a Purcell factor in the strong coupling regime seems somewhat unconventional to me. Is it related to the damping of the Rabi-oscillation?

8) several of the strong coupling cases in the supplement are not convincing. This does not really back the authors main claim...

Reviewer #3 (Remarks to the Author):

The manuscript reports strong coupling between a multilayer stack of the transition metal dichalcogenide WSe₂ and a gold nanogap plasmonic resonator at room temperature. The major claim is that strong coupling with a coupling strength $\gg kT$ is required for some applications, and that the reported results are the first evidence of achieving this regime in TMDCs. The authors also argue that the system is of interest because the small mode volumes in the nanogap cavities are advantageous for producing high nonlinearities, but no experimental measurement of nonlinearity is performed.

The experiments are interesting and the results well presented, but I do not think that enough evidence is presented to support the claim that strong coupling has been observed. In the experimental results in figures 1, 2 and 3, a splitting is clearly seen in the plasmonic scattering spectrum that may indicate strong coupling, but could also be attributed to the presence of a narrow absorption at the TMDC exciton resonance that reduces the intensity of the scattered light at this wavelength.

Furthermore the theoretical condition for strong coupling, that the coupling strength must exceed the loss rates of both oscillators ($g > \kappa, \gamma$), does not appear to be met. The line width of the plasmonic resonance is not stated explicitly, but the Supplementary Information figure S5a indicates a scattering peak of width in excess of 200 meV, larger than the 140 meV splitting observed ($\kappa > g$)

The fitting to the coupled oscillator model for the peak positions alone is insufficient to prove strong coupling. With such a limited range of data other models may provide equally good agreement.

Further evidence is therefore needed to prove strong coupling. This could take several forms:

- 1) Full spectral fits to the scattering spectra in which the exciton and plasmon line widths (γ and κ) appear as fixed parameters to demonstrate that the properties of the coupled system are consistent with those of the individual components,
- 2) Fluorescence spectra that are consistent with a strongly coupled system and the usual rapid relaxation from the upper polariton branch to the lower branch (the PL spectrum in Fig 2e is not commented on in the text but does not appear to be due to polariton emission - the peak position looks the same as in Fig 2d without the plasmonic resonator),
- 3) Scaling of the splitting with the inverse square root of the mode volume or with the square root of the oscillator strength.

Any one of these three additional pieces of evidence may be sufficient to convince that the splitting observed is due to strong coupling.

Two additional corrections are required.

a) In reviewing the current state of the art (lines 42-43) the authors state that so far strong coupling with TMDCs has required cryogenic cooling. In the following sentence they cite works where strong coupling has been observed at room temperature. The first sentence should be reworded to reflect the fact that room temperature strong coupling has been demonstrated.

b) In the same paragraph the authors imply that to clearly resolve polaritons at room temperature

requires a splitting > 100 meV. This number is somewhat arbitrary. Flatten et al (SREP 2016, fig2d,3) demonstrated splittings up to 70 meV, almost three times kT , which are as at least as well-resolved as those in the present work.

In summary I do not think that the manuscript is suitable for Nature Communications in its present form. The experiments are interesting but insufficient evidence is presented that strong coupling has been observed, and there are two counter-indications in the high cavity loss rate and apparently unperturbed PL spectrum. A clear demonstration of strong coupling of the TMDCs in nanogap cavities would be of significant interest to the community due to the potential for stronger nonlinearities, although the increased loss rate of the resonator may limit the usefulness of this approach. The paper could be reconsidered for publication in the journal if substantial additional evidence of strong coupling is presented and these factors are included in the discussion.

Response to referees:

We greatly appreciate the feedback from all 3 referees who note this is a “very interesting and original observation, which in itself merits consideration of publication in a journal like Nat. Comm.” (referee 1), are convinced that our “device is operated in the strong coupling regime, is well justified” (referee 2), and “the reported results are the first evidence of achieving this regime in TMDCs” (referee 3). They raise some concerns, mainly about the behaviour of our system with number of WSe₂ layers and strong coupling regime. Through additional experiments and additional analytical models prompted by these comments, we have been able to address all the issues raised by the reviewers, as discussed in detail below.

Variation of number of TMDC's layers:

- As suggested, we repeated the experiment with different numbers of WSe₂ layers and directly measured the transition from weak to strong coupling regime (Fig. R1).
- The transition from weak to strong coupling is clearly observed at $N_L > 7$.
- We analytically estimated the dipole strength and number of excitons coupling to the system for different numbers of layers of WSe₂. We find a best fit for the exciton interlayer coupling of 5ML.
- Plasmon detuning experiments were now performed also in weak coupling ($N_L=1,4$) and show crossing of exciton line (Fig. S12a) whereas for strong coupling an anti-crossing was observed (Fig. S12b).

Figure R1: **(a)** Measured dark-field scattering spectra for NPoMs with different N_L showing clear splitting for $N_L \geq 7$. **(b)** Bright field images of exfoliated WSe₂ with different layers N_L , confirmed by AFM. **(c)** Analytically calculated z -dipole strength (in Debye) of exciton with increasing N_L for different interlayer coupling lengths δ (see text, solid line 5nm, dashed 2nm and 8nm), maximum dipole contribution is 20% total exciton strength. **(d)** Calculated number of excitons within the mode volume. **(e)** Using the dipole in (c) fits the experimentally obtained Rabi coupling strength. Grey points are from measured plasmon linewidths, where NPoM size is changed to keep the exciton on resonance, giving strong coupling regime as orange shaded region.

- We also have taken additional PL data, which allow us to separate contributions from the NPoM and the surrounding multilayer, see new Fig.2 and below. Light emission is clearly seen at the lower polariton. Finally, we now carefully correlated measurements of reflectance and AFM to better quantify the number of layers in each region of the sample, now corrected throughout.

Reviewer 1:

1) Authors state that the multilayer with its high refractive index reduces the extreme red-shift of the nanogap, such that the plasmonic resonance (and high Purcell factor) coincides with the direct A-exciton of the multilayer favouring the coupling. It is a peculiar coincidence that the balance between expanding volume (reduced coupling) and resonant tuning (enhanced coupling) should find its optimum at 8 multilayers, exactly the case the authors measured. Figure 4c seems to indicate enhanced Purcell factors for z-orientation even beyond 8 layers, however the reverse for x-orientation. So is the optimum at 8 or beyond? Why did authors not vary the multilayer thickness experimentally? Data on different thicknesses would certainly strengthen the argumentation.

As suggested, we now show extra data for different numbers (N_L) of multilayers (see above). There is indeed a delicate balance between number of layers, nanoparticle size (to tune the resonance to the exciton) and dipole orientation, to achieve strong coupling. To optimise, for each N_L different nanoparticle size is needed to match the exciton, however for larger N_L irregularities in NP size and shape make this systematic tuning more difficult, while the linewidth also increases thus smearing out strong coupling for $N_L > 15$. A detailed discussion and Fig.4d is added.

2) The direct experimental comparison of a monolayer and a bilayer should be most instructive, as volume and resonance are very similar while the difference in WSe2 bandstructure would directly dominate. Is strong coupling predicted for the bilayer case?

From our calculations it is hard to achieve strong coupling for $N_L < 4$ hence this interesting comparison is not viable. Our experiments agree with this trend, and the emerging small dip only gives clear Rabi splitting of $\hbar\Omega \geq 100$ meV resolved for 7 layers and more. This is now discussed.

3) The strong coupling is clearly observed in the double-peaked dark-field spectra on the multilayer WSe2 constructs. Yet on the exact same system, the A-exciton luminescence is a single fixed peak without any double-peak signature. Why? Is this due to x-orientation, perpendicular to the gap field, of the A-exciton?

The PL spectra previously in Fig. 2d,e showed emission of the multilayer layer film and not from the NPoM construct. We now performed additional PL measurements in which we were able to separate contributions from the bare multilayer, and that from the NPoM gap. We find indeed that the emission in the NPoM is at the lower polariton, as expected. Upper polariton emission is not seen (as for many strong coupling systems) because energy relaxation is too fast. We provide a new Fig.2d,e and discuss this in the paper.

4) Abstract reads: "Strong coupling is impossible with monolayers" and indeed a single plasmon peak is seen. Yet comparing 8 and single layer, the volume V changes around 10x while moving out-of-resonance changes the enhancement by $\sim Q$. Off-resonance balances cavity volume. Thus Q/V is roughly similar for both 8 and single layer. Then the in-plane exciton dipole orientation for the monolayer must be the main reason for the weaker coupling? Or is the delocalization of the exciton even more important? Could authors quantify these factors and estimate the resulting coupling for the monolayer case. Is it still in the meV range or really much lower??

For 1ML the dipole is only in-plane, but as N_L increases indeed the z-dipole contribution increases steeply (Fig.S11). The delocalisation of the exciton between layers leads to the increasing Rabi splitting for the s_{02} mode, as now shown experimentally. We show interlayer coupling at around 5ML fits out data best. We note that delocalised excitons cannot couple to the s_{11} mode because they have opposite symmetry. From our experimentally-observed emission enhancement when tuning through resonance on single layers we estimate $P_F > 2500$ (PL emission in SI Fig.S5) which can be used to extract a coupling of 30meV.

5) Figure 1e and 1f. It would be clearer to swap 1e and 1f such as to have the x orientation side-by-side to the monolayer and the z with the multilayer.

As suggested we swap Figures 1e,f to emphasize the correlation between the number of layers and the orientation of the maximum field enhancement.

Reviewer 2:

1) The authors' plasmonic device, operated in the strong coupling regime with a single emitter, has recently been published in Nature 535, 127 (2016). The room temperature strong coupling with a plasmonic device with 2D materials has been discussed in Liu et al. Nano Letters 16, 1262 (2016). While I think that the paper by the authors is elegant, due to the above mentioned two reports (and the absence of new, other physics), I think it's suited in a more specialized journal.

We believe that the discussions by the other referees strongly make clear the new physics, which relates to dipole orientation and interlayer coupling (not mentioned in Liu et al), the confinement volume (crucial for nonlinearities, also never previously discussed), and the use of single plasmonic constructs (unlike Liu et al who combine effects over many hundreds of localised sites) for room temperature strong coupling of TMDs. The general analysis and summary are clearly of wide interest for the van der Waals materials community, providing physically-backed guidance for designing new devices.

2) Ref. 14 does not seem to be the most appropriate reference in the context of 'low threshold polariton lasers'

We thank the referee for pointing this out and changed the reference.

3) When the authors write 'often spectral [44] splittings [17–19] are barely resolved and are on the order of thermal energies', do they want to give credit that in ref 17-19 spectral splittings were observed, or do they want to express their criticism about the poor visibility about the latter? I think this formulation is very indirect...

We emphasize that reports of Rabi splitting at room temperature show Rabi splittings on the order of thermal energies. To avoid the confusion highlighted we now clarified this.

4) I believe the statement " maximising the Rabi coupling, $\Omega = \sqrt{g^2 - (\gamma_x - \kappa)}$ by minimising $\gamma_x - \kappa$ is not fully correct. The measurable splitting (in reflection, absorption, PL or transmission) follows different formulas which are all detailed in Savona et al. Solid State Communications 93, 733 (1995). This also makes the next statement about the 'visibility' unnecessary. As the authors imply with this statement, increasing cavity damping to compensate emitter damping does not work...

We thank the referee for spotting this, and remove it as suggested.

5) As a side remark, it think it's interesting that the paper by Liu et al. Nature Materials (2015) is only cited in this context, rather than in the introduction as the first claim of strong coupling with a monolayer of MoS₂. I think either the authors should give full credit to the paper as the first strong coupling demonstration in TMDCs, or fully remove the article from the reference list.

We thank the referee for spotting this, and amend as suggested.

6) I think the discussion about the nonlinearity parameter is of pedagogical value, but not backed by any experimental data as far as I can see. As such, it does not significant value to the paper in my opinion.

Indeed we emphasise this pedagogical idea, but it is important since the real interest in such strong coupled systems is less the linear than the nonlinear domain. Our explanation suggests why nonlinear features are hard to see, compared to extensive work in III-V and GaN semiconductor polaritons (that we have worked on extensively). We also now show nonlinearities in this system, which shows clear nonlinear emission. We believe this discussion helps guide future work.

7) I could not track the references in Table 8.

We added a bibliography to the supplementary Information to make the references more accessible.

8) The concept of a Purcell factor in the strong coupling regime seems somewhat unconventional to me. Is it related to the damping of the Rabi-oscillation?

The Purcell factor is used here as a figure of merit to describe the system across both domains, and has been much investigated so is a useful point of connection to previous work. Even in the strong coupling domain, it is well defined, although does not describe the subsequent Rabi oscillations.

9) Several of the strong coupling cases in the supplement are not convincing.

We provide a wide range of typical data to aid the reader, all of which shows clear strong coupling.

Reviewer 3:

1) The manuscript reports strong coupling between a multilayer stack of the transition metal dichalcogenide WSe₂ and a gold nanogap plasmonic resonator at room temperature. The major claim is that strong coupling with a coupling strength $\gg kT$ is required for some applications, and that the reported results are the first evidence of achieving this regime in TMDCs. The authors also argue that the system is of interest because the small mode volumes in the nanogap cavities are advantageous for producing high nonlinearities, but no experimental measurement of nonlinearity is performed. Do we need this non-linearity thing?

As discussed above (referee 2 Q6), we believe the nonlinearity is the critical *raison d'être* of strong coupling work, and better understanding the different geometries is vital to progress this area. We now provide first experiment evidence of such nonlinearities.

2) The experiments are interesting and the results well presented, but I do not think that enough evidence is presented to support the claim that strong coupling has been observed. In the experimental results in figures 1, 2 and 3, a splitting is clearly seen in the plasmonic scattering spectrum that may indicate strong coupling, but could also be attributed to the presence of a narrow absorption at the TMDC exciton resonance that reduces the intensity of the scattered light at this wavelength.

In the paper, we compare directly anti-crossings for different set of NPoMs as well as in the SI we show the case of weak coupling where a dip in the scattering is seen, but no anti-crossing while tuning [Fig. S12]. To further convince the referee, we now show the layer dependence of the Rabi splitting [Fig. 4d, S11] which explicitly shows the strong coupling condition.

3) Furthermore the theoretical condition for strong coupling, that the coupling strength must exceed the loss rates of both oscillators ($g > \kappa, \gamma$), does not appear to be met. The line width of the plasmonic resonance is not stated explicitly, but the Supplementary Information figure S5a indicates a scattering peak of width in excess of 200 meV, larger than the 140 meV splitting observed ($\kappa > g$)

Directly extracting $\kappa=90\text{meV}$ from Figure 1a and $\gamma_x=25\text{meV}$ from Figure S2a, the strong coupling condition $2g > \kappa + \gamma_x$ is satisfied for $2g > 115\text{meV}$. This is also seen in the layer dependence of

Fig.R1. The linewidth used by the referee from Figure S5a shows that by contrast the plasmon resonance of Ag cubes is in the weak coupling regime since the strong coupling condition indeed cannot be not fulfilled with $2g > 30\text{meV}$. We also add this comparison as the new Fig.4d.

4) The fitting to the coupled oscillator model for the peak positions alone is insufficient to prove strong coupling. With such a limited range of data other models may provide equally good agreement.

Instead of relying on fitting the coupled oscillator model to the peak positions, we directly show the anti-crossing by tuning the plasmon. We also now show the splitting depends drastically on the number of layers, as expected for the strong coupling regime. Finally we also now show the fluorescence spectra track the strong coupling polaritons.

Further evidence is therefore needed to prove strong coupling. This could take several forms. Any one of these three additional pieces of evidence may be sufficient to convince that the splitting observed is due to strong coupling.

As noted below, we address not one, but all these additional pieces of evidence:

a) Full spectral fits to the scattering spectra in which the exciton and plasmon line widths (γ and κ) appear as fixed parameters to demonstrate that the properties of the coupled system are consistent with those of the individual components,

As suggested we made this fit (see Fig.R2), and now added it in the SI as Fig.S11.

Figure R2: Comparison of two coupled oscillator model for linewidths (solid lines) to extracted experimental data (points). The FWHM linewidths are extracted by fitting Lorentzian peaks to the experimental data from Fig.3(e). The model extracts the linewidths for lower (Γ_-) and upper (Γ_+) polaritons, using the same polariton coupled oscillator model as in the main text, using fixed linewidths for the plasmon and exciton. In the negative detuning case (plasmon below exciton energy) we also expect the plasmon radiative coupling to increase (see ref [24] in main text).

b) Fluorescence spectra that are consistent with a strongly coupled system and the usual rapid relaxation from the upper polariton branch to the lower branch (the PL spectrum in Fig 2e is not commented on in the text but does not appear to be due to polariton emission - the peak position looks the same as in Fig 2d without the plasmonic resonator),

As discussed above (referee 1 Q3), we now clearly observe the PL of the NPoM-WSe₂ strong coupled system and find that the emission is as expected on the lower polariton branch. We discuss this in more detail in the text.

c) Scaling of the splitting with the inverse square root of the mode volume or with the square root of the oscillator strength.

Changing the number of layers changes $g(\mu_{ex}, n_{ex}, V_m)$ because all three parameters correspondingly tune at the same time. However it is therefore not possible to plot g against just mode volume or oscillator strength independently. We carefully take all three parameters into account to show the experimental g indeed matches the strong coupling analytical model (Fig.R1).

8) In reviewing the current state of the art (lines 42-43) the authors state that so far strong coupling with TMDCs has required cryogenic cooling. In the following sentence they cite works where strong coupling has been observed at room temperature. The first sentence should be reworded to reflect the fact that room temperature strong coupling has been demonstrated.

Indeed this is helpful and we changed the sentence accordingly.

9) In the same paragraph the authors imply that to clearly resolve polaritons at room temperature requires a splitting > 100 meV. This number is somewhat arbitrary. Flatten et al (SREP 2016, fig2d,3) demonstrated splittings up to 70 meV, almost three times kT , which are as at least as well-resolved as those in the present work.

We meant this to refer to plasmonic cavities, which require >100 meV splitting due to plasmonic damping of 90meV. We clarified this now in the text, and show the comparison in Fig.4d.

As the referees emphasise, we are thus convinced that the fundamental insights provided by this work will draw the interest of the wide scientific community and certainly appeal to the interdisciplinary readership of Nature Communications.

Reviewers' comments:

Reviewer #1 (Remarks to the Author):

Kleemann and coauthors have revised their manuscript, taking into account quite satisfactory all issues raised.

As to the novelty. Indeed the methodology is based on the "NPoM" nanoparticle on mirror geometry, as published by the group Nature 535, 127 (2016) for strong coupling of a single organic molecule in a picocavity. Current manuscript shows first application of single site NPoM geometry to a 2D-TMD semiconductor. Strong coupling of a monolayer of MoS₂ (not WSe₂) was published by Liu et al. in NanoLetters 16, 1262 (2016); yet this work finds 58 meV, at low temperature 77K and on a macroscopic arrays of nanodisks. That presented nanodisk field confinement is much less, while Liu et al. still claim strong coupling even for a monolayer. As such that Liu work seems not really consistent with the current manuscript presented by Kleeman et al. and would be good to discuss that difference. In any case, Kleeman presents results at room temp, in single cavity sites, now with systematically varied number of TMD layers, and addresses the orientation, thus goes substantially beyond the Liu work in insight gained.

It is very nice that authors went through the exercise to collect extra data for varying number of multilayers and thus reveal the trend and optimum in trade-off between number of layers, resonance and dipole orientation for strong coupling. Moreover, it made me realize that for every layer thickness a suitable nanoparticle size is needed to keep the resonance. All very insightful, and nice to see the new data in the main manuscript, making the story stronger.

My question on the single peak PL spectrum got answered by new data, again, with PL spectra both from the bare multilayer, and from the NPoM gap, showing emission of the NPoM only at the lower polariton. Good to see this clarified in the main manuscript in new Fig.2d,e, which addresses fully my earlier comments.

It is interesting that for the monolayer case, despite weaker coupling, authors still guess a coupling of about 30 meV, not strong, but not that weak either. Would be insightful to contain that info in the main manuscript. This also in connection with the Liu manuscript who claim "strong coupling" of 58 meV at 77 K for their MoS₂ monolayer under a nanodisk.

Reviewer #2 (Remarks to the Author):

I thank the authors for resubmitting their work. However, my main criticism persists, as their main result, in my point of view, is the demonstration of another platform to observe room temperature strong coupling with a layer of TMDC.

I disagree with the authors when they imply that their device facilitates a much better resolution of Rabi-splittings as in the previous works, in fact, the order of magnitude for the visibility of the doublet peaks is well within the same order of magnitude, with a substantial amount of noise on top of the data.

While I believe that the observation of a non-linearity (polariton lasing?) would make a strong difference, it becomes clear from the way these data are presented and discussed, that this piece of the work still is pre-mature and certainly deserves further investigation (such as details on energy shifts, narrowing on linewidth, coherence...).

Therefore, my previous conclusion holds, and I do not recommend to publish this paper in its present form in Nature Communications.

Reviewer #3 (Remarks to the Author):

My concerns regarding the demonstration of strong coupling have been addressed thoroughly in the revisions. I am happy for the paper to be published in Nature Communications in its current form.

Response to the Referees:

Reviewer 1:

1) Strong coupling of a monolayer of MoS₂ (not WSe₂) was published by Liu et al. in NanoLetters 16, 1262 (2016); yet this work finds 58 meV, at low temperature 77K and on a macroscopic arrays of nanodisks. That presented nanodisk field confinement is much less, while Liu et al. still claim strong coupling even for a monolayer. As such that Liu work seems not really consistent with the current manuscript presented by Kleeman et al. and would be good to discuss that difference. In any case, Kleeman presents results at room temp, in single cavity sites, now with systematically varied number

of TMD layers, and addresses the orientation, thus goes substantially beyond the Liu work in insight gained.

As the reviewer points out, the system used by Liu et al. is quite different to our system. There are three major differences that enable Liu et al. to reach strong coupling with a Rabi splitting of 58meV at 77K although they have less confinement.

1. Mode volume:

They use arrays of silver nano disks which support long-range lattice diffraction modes and localized dipolar modes with larger mode volumes than our nanoparticle on mirror cavity. We reach mode volumes as small as 40nm^3 by coupling to the gap modes that are tightly confined. Liu et al. show strong coupling for arrays of nano disks of 100nm diameter, with active volumes more than a thousand times larger.

2. Number of excitons:

Due to the higher mode volume, the number of coupled excitons increases ($g \propto \sqrt{N/V}$),

Therefore increasing the overall oscillator strength. Furthermore, as Liu et al. state in their paper: "Although LSPRs are highly localized, the presence of long-range lattice diffraction modes leads to coherent coupling of the MoS₂ exciton-LPSR "plexitonic" system at different regions, much beyond the length scales of the localized plasmons or excitons thereby increasing the oscillator strength of various coupled resonances." Nonetheless the increase in mode volume dominates over the increase in oscillator strength resulting in a smaller overall coupling constant.

3. Field and exciton dipole alignment:

For our NPoM geometry more than four layers are needed to reach strong coupling, but this is not the case for Liu et al. Their dipolar mode while less confined is oriented in plane with the monolayer excitons of WSe₂ allowing for coupling even in the monolayer case.

We thus now provide a brief discussion of this comparison in the manuscript.

2) It is interesting that for the monolayer case, despite weaker coupling, authors still guess a coupling of about 30 meV, not strong, but not that weak either. Would be insightful to contain that info in the main manuscript. This also in connection with the Liu manuscript who claim "strong coupling" of 58 meV at 77 K for their MoS₂ monolayer under a nanodisk.

As suggested by the reviewer we now add a comment in the manuscript. As discussed in (1) above, the alignment of the field with the orientation of the excitons is crucial to reach the strong coupling regime, requiring out-of-plane oscillator strength in our NPoM case.

Reviewer 2:

1) I disagree with the authors when they imply that their device facilitates a much better resolution of Rabi-splitting as in the previous works, in fact, the order of magnitude for the visibility of the doublet peaks is well within the same order of magnitude, with a substantial amount of noise on top of the data.

We disagree with the reviewer. Our resolution of Rabi splitting at 300K is clearly larger than the line-width of the cavity and emitter losses. The signal to noise is a function of measurement time, and doesn't affect the observed coupling strength and dynamics of strong coupling.

2) While I believe that the observation of a non-linearity (polariton lasing?) would make a strong difference, it becomes clear from the way these data are presented and discussed, that this piece of the work still is pre-mature and certainly deserves further investigation (such as details on energy shifts, narrowing on linewidth, coherence...).

We report clearly the results of the nonlinearity observed (which the referee requested). It is evident from Fig.2d that energy shifts are much smaller than the emission linewidth, and that linewidth

narrowing is also not strongly seen. This is not polariton lasing. We thus report intriguing results topical to the community, while we agree indeed further detailed studies are of interest.

We believe our manuscript is clearly improved by the additions made, as the other two referees confirm. Apart from the demonstration that we can reach the strong coupling regime by incorporating WeSe_2 multilayer stacks, we show that our system is highly tuneable and therefore adjustable to different 2D materials. We provide detailed insights to exciton-plasmon coupling dynamics and show evidence of nonlinear emission paving the way for room temperature polaritonic devices. We thus feel it is highly suited to expedited publication in Nature Communications.

REVIEWERS' COMMENTS:

Reviewer #1 (Remarks to the Author):

Upon revision authors have satisfactorily answered all my questions.

I am happy to see the comparison with the Liu NanoLetters paper contained in the revision, which puts the current paper in perspective and more clearly shows its novelty in achieved cavity volume, control and coupling strength, all at 300K.

As to the criticism of referee 2 on the resolution of the observed Rabi-splitting, I must say presented splitting data is quite convincing and indeed dominant over cavity quality and emitter losses.

I see no need for further iterations on this manuscript.